# Regular Sport Activity Is Able to Reduce the Level of Genomic Damage

**DOI:** 10.3390/biology12081110

**Published:** 2023-08-09

**Authors:** Alfredo Santovito, Angiolina Agostinovna Nigretti, Amedeo Sellitri, Manuel Scarfò, Alessandro Nota

**Affiliations:** Department of Life Sciences and Systems Biology, Via Accademia Albertina 13, 10123 Turin, Italy; angiolinaagost.nigretti01@universitadipavia.it (A.A.N.); amedeo.sellitri@edu.unito.it (A.S.); manuel.scarfo@gmail.com (M.S.); alessandro.nota@conted.ox.ac.uk (A.N.)

**Keywords:** sport, micronuclei, gene polymorphisms, buccal mucosa cells

## Abstract

**Simple Summary:**

Intense physical activity can induce oxidative damage to cells, tissues and genomic material. In contrast, regular but moderate exercise was found to generate lower concentrations of free radicals, as a result of a favorable adaptive response by the organism. We evaluated, with the buccal micronucleus assay, the level of genomic damage in a sample of amateur athletes engaged in different disciplines. We compared the obtained data with those of subjects who practice sports only occasionally and subjects who do not practice sport at all. The aim of the study was to evaluate whether physical activity affects background levels of genomic damage, and whether the different sports disciplines induce varying levels of impact. Furthermore, our aim was to evaluate the role of some polymorphisms of gene-encoding enzymes belonging to the different damage repair systems and metabolic genes in differentially affecting these levels of DNA damage. Athletes showed significantly lower values of micronuclei, nuclear buds and binucleated cells with respect to controls. Among athletes, Sprinters and Martial Artists showed significantly higher frequencies of micronuclei than other categories. Finally, neither sex nor genetic polymorphisms seemed to influence the levels of genomic damage, further confirming that the observed genomic damage is probably due to the nature of the sport activity.

**Abstract:**

Regular physical activity is considered one of the most valid tools capable of reducing the risk of onset of many diseases in humans. However, it is known that intense physical activity can induce high levels of genomic damage, while moderate exercise can elicit a favorable adaptive response by the organism. We evaluated, by the buccal micronuclei assay, the frequencies of micronuclei, nuclear buds and binucleated cells in a sample of amateur athletes practicing different disciplines, comparing the obtained data with those of subjects who practiced sports just occasionally and subjects that did not practice sport at all. The aim was to evaluate whether physical activity affects background levels of genomic damage and whether the different sports disciplines, as well as some gene polymorphisms, differentially affect these levels. A total of 206 subjects, 125 athletes and 81 controls, were recruited. Athletes showed significantly lower values of micronuclei, nuclear buds and binucleated cells with respect to controls. Sprinters and Martial Artists displayed significantly higher frequencies of micronuclei than other categories of athletes. Finally, neither sex nor gene polymorphisms seemed to influence the levels of genomic damage, confirming that the observed genomic damage is probably due to the nature of the sport activity.

## 1. Introduction

In humans, regular physical activity is considered one of the most valid tools able to reduce the risk of premature mortality and onset of many diseases, such as obesity, diabetes, cancer and metabolic and cardiovascular diseases [1,2]. Some of these diseases have been associated with DNA damage, including aberrant DNA methylation patterns and telomere shortening [2,3]. Epidemiological studies have shown that subjects who exercise regularly have a lower risk of all-cause mortality with respect to sedentary subjects [1,4], with beneficial effects that are more pronounced in elderly people [5]. Compared to inactive individuals, physically active adults exhibit improved cardiorespiratory fitness and muscle strength, cognitive function and favorable metabolic profile, as well as healthier body mass and composition [1,3].

At the cellular level, physical activity can determine nuclear changes, influencing gene expression and, in particular, inducing epigenetic modifications in terms of altered DNA methylation in muscle cells [6]. These epigenetic changes induced by the exercise were also found to have beneficial effects in cancer patients, inducing the increase in tumor suppressor gene expression and decreased expression of oncogenes [7]. In particular, it was observed that cancer cells exhibit an abnormal DNA methylation pattern, such as hypermethylation of tumor suppressor gene promoters and hypomethylation in promoter regions of oncogenes [1]. Vice versa, physical activity seems able to reduce the consequences of this process and to determine increased levels of tumor suppressor gene expression and the inhibition of oncogenesis processes [7]. Moreover, physical exercise acts as an inhibitor of the aging process, through the activation of telomerase, the preservation of telomere length [2,6] and the improvement of mitochondrial biogenesis and function [8].

Although it is known that exercise improves health and has a protective effect against diseases, it increases the production of reactive oxygen species (ROS) and free radicals, as part of the muscular adaptation to the training process [9]. Notably, it is known that intense physical activity, combined with some lifestyles (e.g., smoking habits, alcohol intake, incorrect diet) or environmental factors (e.g., exposure to radiation, viruses and bacteria) is able to induce high concentrations of free radicals and ROS, determining inflammation, increased levels of oxidative damage to cells and tissues and genomic damage [10,11]. Furthermore, varying concentrations of some metals, such as iron and copper, can also generate radicals [12]. From a genomic point of view, excessive ROS production is associated with oxidative DNA damage and cancer progression [13]. This can happen, for example, because chromatine-bound iron reacts with environmental H_2_O_2_, producing OH radical, a very reactive species that affects nearby DNA [12].

On the contrary, regular but moderate exercise has been associated with lower levels of free radicals, as a result of a favorable adaptive response by the organism, resulting in beneficial effects in relation to the onset and progression of various pathologies associated with oxidative stress [14]. In particular, research conducted on both humans and rodents has shown that low levels of oxidative stress can promote adaptation and activate cellular mechanisms of protection [14,15].

Based on these assumptions, we decided to evaluate, using the buccal micronucleus (MNi) assay, the level of genomic damage in a sample of amateur athletes practicing different disciplines, and to compare the obtained data with those of subjects who practice sports only occasionally and subjects who do not practice any physical activity.

MNi analysis in exfoliated buccal cells is a useful and non-invasive method to monitor genomic damage in humans, as a consequence, for example, of exposure to genotoxins, radiation, chemicals and environmental xenobiotics [16]. MNi derive from whole chromosomes or chromosome fragments that, during mitotic telophase, are not included in the daughter nuclei due to lack of proper attachment to the spindle during the segregation process in anaphase [17]. These chromosomes or chromosome fragments are then enclosed by a nuclear membrane and, except for their smaller size, are morphologically similar to nuclei [16].

Another nuclear abnormality observable with the MNi test, known as Nuclear Bud (NBUD), has been associated with chromosomal instability events. Nuclear buds are the result of an abnormal gene amplification process and share a similar morphology with MNi, except that they are connected to the nucleus by a narrow or broad stem of nucleoplasmic material. The MNi assay also allows one to record the presence of binucleated cells, the excess of which may represent the result of an imperfect cytodieresis mechanism [17,18,19]. Research examining the impact of sport activity on the frequency of MNi and other aberrations is limited and usually relies on a relatively small samples sizes [20,21,22]. While some studies revealed a correlation between physical activity and an increase in MNi frequency [21], particularly when intense [20], others did not find a significant effect [22]. It is essential to conduct more extensive investigations in this area to better understand the possible relationship between sport activity and these biomarkers.

Finally, it is known that some metabolic and DNA-repair gene polymorphisms are able to modulate the level of MNi and NBUDs [23,24]. For example, the mutated allele for *CYP1A1* exon 7 (A > G) and the *GSTT1* positive and *GSTM1* positive genotypes were found to be associated to variations in the frequency of chromosomal aberrations [23]. Similarly, mutations in *XRCC1* and *XPC* DNA-repair gene polymorphisms were found to be associated with increased levels of cytogenetic damage, such as chromosomal aberrations and sister chromatid exchanges [23,24].

Therefore, we also decided to evaluate the association of some polymorphisms of gene encoding enzymes belonging to the different damage repair systems and metabolic genes with the MNi frequency observed in the studied samples. In particular, we analyzed the following polymorphisms of phases I and II metabolic genes and damage repair genes, which are the most studied gene polymorphisms associated to cytogenetic damage [20]: Cytochrome P450 1A1 (*CYP1A1*) exon 7 (A > G, rs1048943), the Glutathione S-transferases (*GSTT1*) (positive > null, rs1601993659) and GS*TM1* (positive > null, rs1183423000), *XRCC1* 194 (C > T, rs1799782) and *XPC* exon 15 (A > C, rs2228001).

The aims of the present study were:(a)to evaluate whether physical activity affects background levels of MNi, NBUDs and binucleated cells;(b)to test whether the different sports disciplines differentially affect these levels;(c)to evaluate the possible influence of some gene polymorphisms on the frequencies of the analyzed genomic markers.

## 2. Materials and Methods

### 2.1. Population Sample

The study population included 125 athletes (engaged in the following disciplines: Martial Arts, n = 34, Basketball, n = 29, Volleyball, n = 40 and Sprint, n = 22) and 81 control subjects (50 sedentary controls and 31 sports controls, i.e., subjects who practice sports occasionally, and no more than twice a week). Martial Arts, Basketball and Volleyball are considered intermittent metabolic activities, i.e., sports where aerobic activity is alternated with anaerobic activity and breaks. Vice versa, sprint activity is considered an anaerobic sport. For all sports, the training sessions were 1–2 h long, 3–6 times/week.

The control group was subdivided in two groups: (a) “sedentary controls”, that included subjects who do not practice any sport activity at all, and (b) “sport controls”, a category which included subjects who practice sport only occasionally, no more than 2 times/week and no more than 1.5 h per training.

In order to determine physical activity-related and physiological variables (i.e., age, sex and weight) a questionnaire was provided to all study participants. Weight and height data were recorded on the basis of what was indicated in the questionnaire by each subject participating in the study. However, we would like to emphasize that all athletes, before starting the training session, weighed themselves. Therefore, the data they provided can be considered highly realistic. For the sake of consistency, we adopted the same criteria for the control subjects.

It is well known that cigarette smoke, alcohol consumption, drugs and X-rays can alter level of genomic damage [25,26,27,28]. For this reason, we excluded from the sample smokers and individuals who reported alcohol consumption, treatment for acute infections and/or chronic non-infectious diseases, history of cancer and exposure to diagnostic X-rays, for at least one year prior the analysis. Overall, we excluded a total of 220 subjects.

Sampling began in November 2021 and ended in June 2022.

Subjects received detailed information about the aims and experimental procedures of the study and gave their informed consent. Selected volunteers were anonymously identified by a numeric code. The study was approved by the University of Turin ethics committee (protocol number 0609375, 10-28-2021) and was performed in agreement with the ethical standards laid down in the 2013 Declaration of Helsinki.

### 2.2. Buccal MNi Assay

Buccal MNi assay was performed according to the following protocol: exfoliated buccal mucosa cells were collected with a toothbrush by gently scraping the mucosa of the inner lining of one or both cheeks and/or the inner side of the lower lip and palate. The toothbrush tip was then immersed in a fixative solution consisting of methanol/acetic acid 3:1, shaken for at least 1 min and stored at 4 °C before the analyses. Successively, cells were gathered by centrifugation, the supernatant was discarded and the pellet was dissolved in a minimal amount of fixative, which was seeded on slides to detect MNi by conventional staining with 5% Giemsa (pH 6.8) prepared in a Sörensen buffer. Microscopic analysis was performed at 1000× magnification on a light microscope. According to the established criteria, MNi, NBUDs, and binucleated cells were scored in 1000 cells with well-preserved cytoplasm per subject [23].

### 2.3. DNA Extraction and Genotyping

In order to extract DNA, a second toothbrush was used to collect exfoliated buccal mucosa cells by scraping the mucosa of the inner lining of one or both cheeks. DNA extraction was performed using a salting-out procedure: samples were centrifuged at 14,000 rpm for 1 min and the pellet, constituted by buccal mucosa cells, was resuspended in a solution containing 340 µL lysis buffer (10 mM Tris pH 7.6, 10 mM EDTA and 50 mM NaCl), 30 µL of SDS 10% and 30 µL of 10 mg/mL proteinase K. After incubation at 55 °C for 1 h, 200 µL of saturated sodium acetate was added to the solution. The samples were vigorously shaken and centrifuged at 14,000 rpm for 5 min. Subsequently, 0.6 mL of isopropanol for DNA precipitation was added to the supernatant and, after centrifugation at 14,000 rpm for 1 min, 0.5 mL of 70% ethanol was added to remove salt from the pellet. After subsequent centrifugation, the pellet was dried at room temperature for at least 60 min and then resuspended in 50 µL of ultrapure distilled water.

PCR-based genotyping was performed for the following gene polymorphisms: *CYP1A1* (rs1048943, A > G), *GSTM1* (rs 1183423000, presence/absence), *GSTT1* (rs1601993659, presence/absence), *XRCC1* (rs1799782, C > T) and *XPC* (rs2228001, A > C). Primer sequences, annealing temperatures, PCR methodologies and expected PCR product sizes are reported in Appendix A.

PCR reactions were performed in a 25 μL volume containing approximately 10 ng DNA (template), with a final concentration of 1X Reaction Buffer, 1.5 mM of MgCl_2_, 5% of DMSO, 250 µM of dNTPs, 0.5 μM of each primer and 1 U/sample of Taq DNA polymerase (Fischer Scientific Italia, Segrate (MI), Italy). Cycles were set as follows: 35 cycles, 1 min at 95 °C, 1 min at 60–65 °C, 1 min at 72 °C and a final extension step of 10 min at 72 °C. Amplification products were detected by ethidium bromide staining after 2.5% agarose gel electrophoresis.

### 2.4. Statistical Analysis

All statistical analyses were performed using the SPSS software statistical program (version 28.0, SPSS Inc., Chicago, IL, USA). Counts of micronuclei and other abnormalities are presented as the mean frequency (±standard deviation) in a sample of 1000 cells/subject. The distribution of genomic markers was tested for normality using the Shapiro–Wilk test. For not-normally distributed variables, we used the non-parametric Mann–Whitney test, whereas for normally distributed variables the ANOVA test was used. An χ^2^ test contingency table was used to evaluate the Hardy–Weinberg equilibrium (HWE). The statistical differences between Athlete and Control groups in terms of MNi, NBUDs and binucleated cells, as well as the possible association between gene polymorphisms and these genome damage markers were evaluated by Mann–Whitney test. All *p*-values were two-tailed, and the level of statistical significance was set at *p* < 0.05 for all tests.

## 3. Results

Results of the Shapiro–Wilk test showed that all the analyzed categories, with the exception of the weight (*p* = 0.388), were not normally distributed (*p* < 0.001).

In order to calculate the statistical significance of the obtained results, we performed the power analysis. The results of power analysis showed a very high power value of our samples (0.99) for all biomarkers, with exception of binucleated cells for the “sport controls” group, showing a power value of 0.65.

The demographic features of the studied samples are reported in Table 1. A total of 206 subjects were recruited, 148 males ($\bar{X}$ ± σ, age: 21.858 ± 3.720; weight: 74.418 ± 10.081) and 58 females ($\bar{X}$ ± σ, age 21.052 ± 2.781; weight: 60.716 ± 9.942). As expected, significant differences were found between sexes in terms of means weight (*p* < 0.001, Mann–Whitney test). The athlete sample was represented by 125 subjects ($\bar{X}$ ± σ, age: 21.528 ± 4.063; weight: 72.654 ± 12.103), 89 males ($\bar{X}$ ± σ, age: 21.697 ± 4.342; weight: 76.526 ± 10.496) and 36 females ($\bar{X}$ ± σ, age: 21.111 ± 3.293; weight: 63.083 ± 10.470). Among the athletes, 35 were Martial Arts athletes ($\bar{X}$ ± σ, age: 22.500 ± 4.460; weight: 70.941 ± 9.739); 29 were Basketball players ($\bar{X}$ ± σ, age: 20.207 ± 3.668; weight: 81.652 ± 9.992); 40 were Volleyball players ($\bar{X}$ ± σ, age: 21.300 ± 4.201; weight: 72.223 ± 12.722) and 22 were Sprinters ($\bar{X}$ ± σ, age: 22.182 ± 3.319; weight: 64.227 ± 9.532). The Basketball athletes showed a significantly higher weight with respect to Martial Artists and Sprinters (*p* < 0.001 for both) as well as to Volleyball athletes (*p* = 0.002).

The control sample was represented by 81 subjects ($\bar{X}$ ± σ, age: 21.790 ± 2.375; weight: 67.327 ± 10.512), 59 males ($\bar{X}$ ± σ, age: 22.102 ± 2.524; weight: 71.237 ± 8.563) and 22 females ($\bar{X}$ ± σ, age: 20.955 ± 1.704; weight: 56.841 ± 7.763).

A total of 206,000 buccal cells were observed.

No significant differences were found between sexes in the frequencies of MNi, NBUDs and binucleated cells (Table 2, Figure 1).

In Table 3 and Figure 2, results regarding the statistic evaluation of the differences in the frequencies of MNi, NBUDs and binucleated cell level are shown. In Figure 3, some examples of cells with MNi, NBUDs and binucleated cells are illustrated. Athletes showed significantly lower values of MNi (*p* < 0.001), NBUDs (*p* = 0.002) and binucleated cells (*p* < 0.001) (Table 3) with respect to control subjects.

In order to evaluate the existence of possible differences within the control group, the latter was subdivided into “sedentary controls”, which included subjects who do not practice any sport activity (not even occasionally), and “sport controls”, a category that includes subjects who only practice sport occasionally, no more than 2 times/week and no more than 1.5 h per training (Table 4 and Figure 4). With regard to the analyzed markers, the group of athletes showed significantly lower MNi than both the sports and sedentary controls (*p* < 0.001 for both), whereas for NBUDs, the significance was observed only with respect to the sport controls (*p* < 0.001). No significant differences were found in terms of binucleated cells. Interestingly, the group of sports controls showed significantly higher values of MNi and NBUDs with respect to the sedentary controls (*p* = 0.040 and *p* < 0.001 for MNi and NBUDs, respectively).

In Table 5, the group of athletes was subdivided based on the different sport activity practiced, and the differences in the frequencies of MNi, NBUDs and binucleated cells were statistically evaluated. Sprinters and Martial Artists showed a significantly higher frequency of MNi (*p* = 0.010) and binucleated cells (*p* < 0.001), respectively, with respect to the other athletes categories, whereas the Volleyball athletes showed a significantly higher NBUDs frequency with respect to the Basketball athletes (*p* = 0.030).

Finally, in order to analyze the possible influence of some metabolic and DNA-repair genes on the level of genomic damage, an association analyses between some phase I, phase II and DNA-repair genes and the analyzed genomic damage markers was performed. All the analyzed gene polymorphisms were in the Hardy–Weinberg equilibrium (Table 6). In the whole sample, no association was found between gene polymorphisms and the frequency of MNi, NBUDs and binucleated cells (Table 7).

## 4. Discussion

It is known that intense physical activity is associated with the production of high levels of free radicals, which can deplete the non-enzymatic antioxidant system, inducing impaired cellular function, apoptosis, necrosis and genomic damage [11,29]. On the other hand, the production of free radicals induced by moderate and constant physical exercise is considered one of the most powerful natural stimuli capable of improving the expression of antioxidant enzymes. In fact, trained subjects showed a higher number of mitochondria, with consequent lower levels of respiratory activity, oxidative stress and chronic inflammation than untrained subjects [4]. In general, moderate and regular physical activity was found to preserve genomic integrity and tissue function and to reduce the occurrence of age-related chronic diseases [3]. Moreover, sport activity is a potent stimulator of muscle protein synthesis: it was demonstrated that the levels of muscle protein synthesis after acute exercise are modulated by an individual’s training status, whereas stress induced by an unaccustomed resistance exercise, typical, for example, of subjects who practice sports only occasionally, could result in muscle damage [30].

In the present study, we evaluated the possible influence of moderate and constant physical activity on the levels of genomic damage, also comparing the frequencies of analyzed genomic markers between the different groups of athletes.

Athletes showed a significantly lower frequency of MNi than controls (Table 3), demonstrating that regular physical exercise is able to reduce the level of this type of genomic damage. In fact, the existence of an adaptive response induced by ROS following regular long-term training is known; this is probably the result of an over-expression of antioxidant enzymatic systems, such as superoxide dismutase, catalase and glutathione peroxidase. These endogen antioxidant enzymes act synergistically with non-enzymatic antioxidants, i.e., vitamins C and E, to counterbalance the negative effects of oxidative stress induced by physical exercise [31].

However, the constancy of physical exercise also appears to differently influence the level of oxidative stress [32]. In particular, it has been observed that, while moderate and regular physical exercise induces the upregulation of antioxidant and oxidative damage repair systems [31], acute and occasional physical activity triggers a massive production of free radicals, with a depletion in antioxidant defenses and an increase in oxidative damage to proteins and DNA [33,34,35].

These findings are congruent with our results; in fact, when we compared the group of athletes with both the sedentary and sport controls, we observed that this last group, i.e., subjects who perform sport activities occasionally and often inconsistently and without letting the body develop antioxidant defenses, showed significantly higher MNi values than both sedentary controls and athlete groups (Table 3). This could mean that, with respect to individuals who practice sport regularly, subjects who perform sport occasionally, and thus overtrain their body, could be more prone to genomic damage and, consequently, have an increased risk for some diseases.

It should be noted that the group of sedentary controls did not show significant differences in terms of frequencies of MNi and NBUDs compared to the group of athletes (Table 4). This finding could also be explained by the fact that the number of “background” MNi observable in unexposed young subjects is very low, falling within a range from 0 to 12 MNi observed for 1000 analyzed cells [36] and, therefore, it is unlikely that significant differences will emerge in a comparative analysis.

The next step of the present work was to separate the group of athletes on the basis of the different disciplines (Table 5). Sprinters, subjects practicing an anaerobic activity, showed a significantly higher frequency of MNi with respect to the other athletes. Although aerobic and resistance training was considered the leading cause of oxidative stress, it has been observed that free radicals can also be produced through other pathways, which are not necessarily related to oxygen demand. In fact, several studies have shown that anaerobic exercise (high intensity training) can produce levels of oxidative stress similar to the ones produced with aerobic physical activity [33].

Contrary to what has been observed in some published papers [36,37], sex was not found to influence the frequency of MNi, NBUDs and binucleated cells.

Finally, it is known that *GSTM1* and *GSTT1* gene polymorphisms can affect the levels of exercise-induced oxidative stress [38,39] and that some polymorphisms of damage repair genes can code for enzymes that show greater or lesser efficiency in repairing DNA damage and, in this sense, can affect the level of MNi [23]. In our sample, the gene polymorphisms analyzed do not seem to influence the frequencies of MNi and NBUDs, further confirming that the observed differential genomic damage is probably due to the intensity and type of sport activity. However, in interpreting these data, it is necessary to consider that the response to the sport activity is multifactorial, being influenced by various variables including physiology, metabolism and genetics. From this point of view, it is unlikely that the genetic component can be explained by variation in the DNA sequence of a few genes, while it is more probable that several gene loci, each with a small but significant contribution, could be responsible for this component [31].

## 5. Conclusions

With the present study, we showed, in a sample of amateur athletes, that regular physical exercise can reduce the level of genomic damage measured in terms of MNi and NBUDs frequencies. This result is probably attributable to an over-regulation of the endogen antioxidant systems induced by moderate sport practice, as widely reported in the literature [14,15].

Another relevant result we obtained is that sport controls showed significantly higher MNi values than both the sedentary controls and athletes, demonstrating that practicing sport activity irregularly and inconsistently may even result in an increase in the genomic damage.

Finally, a limitation of the present study is the under-representation of certain categories of athletes, notably endurance athletes. However, we would like to highlight that our selection process ensured that the involved subjects were free of any confounding factor (i.e., smoking, alcohol consumption, radiation exposure, drug intake). These characteristics made it challenging to find suitable subjects.

## Figures and Tables

**Figure 1 biology-12-01110-f001:**
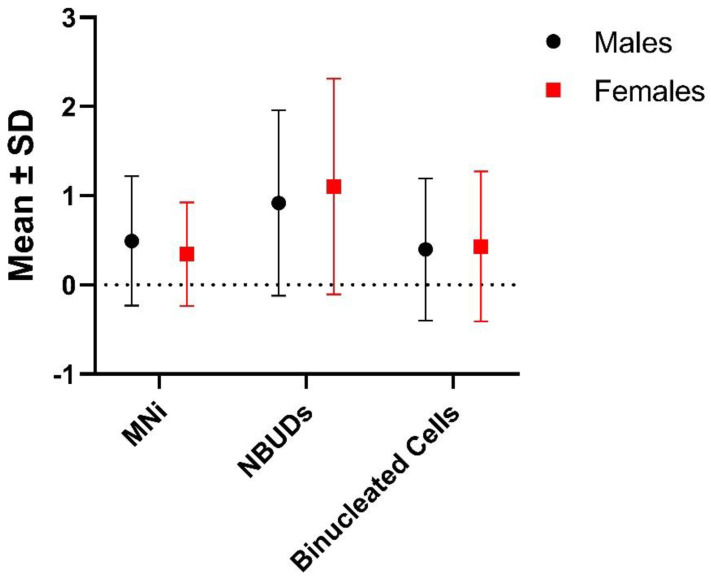
Differences between males and females in the frequencies of MNi, NBUDs and binucleated cells. SD = Standard Deviation.

**Figure 2 biology-12-01110-f002:**
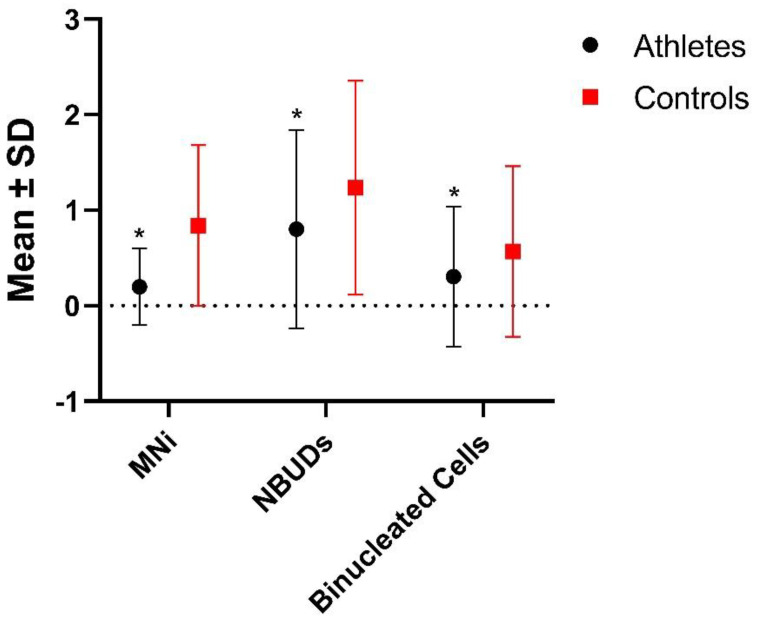
Differences in the frequencies of MNi, NBUDs and binucleated cells between athletes and controls groups. SD = Standard Deviation. * *p* < 0.001 with respect to controls (Mann–Whitney test).

**Figure 3 biology-12-01110-f003:**
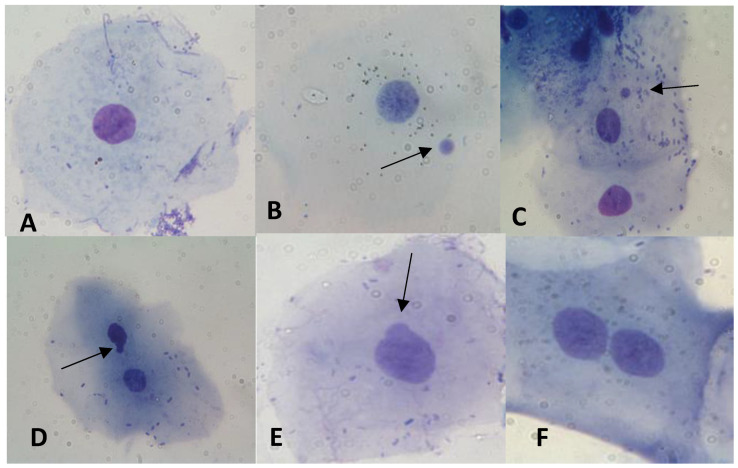
Examples of (**A**) normal cell; (**B**,**C**) cells with MNi; (**D**,**E**) cells showing NBUDs and (**F**) binucleated cells, observed in our samples. Arrows indicate the aberrations.

**Figure 4 biology-12-01110-f004:**
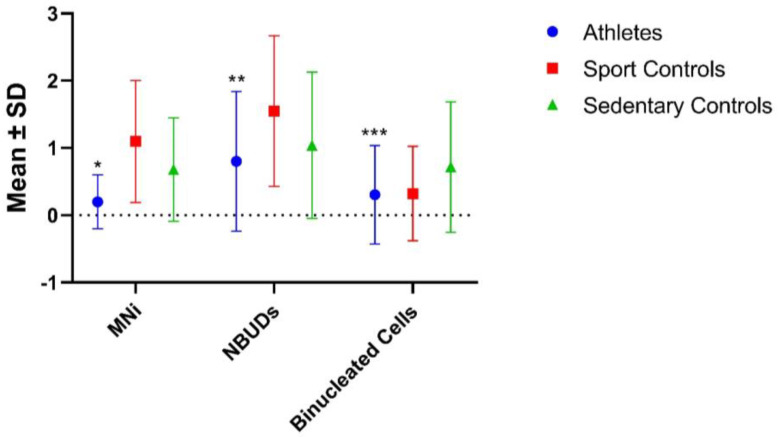
Comparison between the groups of athletes, sedentary controls and sport controls as regards genomic damage. SD = Standard Deviation. * *p* < 0.001, significantly lower with respect to sedentary and sport controls (Mann–Whitney test); ** *p* < 0.001, significantly lower with respect to sedentary and sport controls (Mann–Whitney test); *** *p* < 0.001, significantly lower with respect to sedentary controls (Mann–Whitney test).

**Table 1 biology-12-01110-t001:** General characteristics of the studied samples.

Subjects	N (Frequency)	Age ($\bar{X}$ ± σ)	Weight ($\bar{X}$ ± σ)
Total Subjects	206	21.631 ± 3.494	70.560 ± 11.770
Males	148	21.858 ± 3.720	74.418 ± 10.081 ^A^
Females	58	21.052 ± 2.781	60.716 ± 9.942
Athlete	125	21.528 ± 4.063	72.654 ± 12.103
Males	89	21.697 ± 4.342	76.526 ± 10.496
Females	36	21.111 ± 3.293	63.083 ± 10.470
Martial Arts	34	22.500 ± 4.460	70.941 ± 9.739
Basketball	29	20.207 ± 3.668	81.652 ± 9.992 ^B,C^
Volleyball	40	21.300 ± 4.201	72.223 ± 12.722
Sprinters	22	22.182 ± 3.319	64.227 ± 9.532
Controls	81	21.790 ± 2.375	67.327 ± 10.512
Males	59	22.102 ± 2.524	71.237 ± 8.563
Females	22	20.955 ± 1.704	56.841 ± 7.763

N = number of studied subjects; $\bar{X}$ = mean; σ = Standard Deviation. ^A^
*p* <0.001, significantly higher with respect to females, ANOVA test. ^B^
*p* < 0.001, significantly higher with respect to Martial Arts and Sprinters (ANOVA Test). ^C^
*p* = 0.002, significantly higher with respect to Volleyball (ANOVA Test).

**Table 2 biology-12-01110-t002:** Evaluation of possible differences between males and females in the frequencies of MNi, NBUDs and binucleated cells.

Subjects	N	MNi Nab ($\bar{X}$ ± σ)	NBUDs Nab ($\bar{X}$ ± σ)	Binucleated Cells Nab ($\bar{X}$ ± σ)
Males	148	73 (0.493 ± 0.724)	136 (0.919 ± 1.040)	59 (0.399 ± 0.797)
Females	58	20 (0.345 ± 0.579)	64 (1.103 ± 1.209)	25 (0.431 ± 0.840)

N = number of studied subjects; Nab = Number of aberrations; $\bar{X}$ = mean; σ = Standard Deviation; MNi = Micronuclei; NBUDs = Nuclear Buds.

**Table 3 biology-12-01110-t003:** Statistical evaluation about differences in the level of MNi, NBUds and binucleated cells between athletes and controls groups.

Subjects	N	MNi Nab ($\bar{X}$ ± σ)	NBUDs Nab ($\bar{X}$ ± σ)	Binucleated Cells Nab ($\bar{X}$ ± σ)
Athletes	125	25 (0.200 ± 0.402) ^A^	100 (0.800 ± 1.040) ^B^	38 (0.304 ± 0.732) ^C^
Controls	81	68 (0.840 ± 0.843)	100 (1.235 ± 1.121)	46 (0.568 ± 0.894)

N = number of studied subjects; $\bar{X}$ = mean; σ = standard deviation; Nab = Number of aberrations; MNi = Micronuclei; NBUDs = Nuclear Buds). ^A, B, C^
*p* < 0.001 with respect to controls (Mann–Whitney test).

**Table 4 biology-12-01110-t004:** Statistical comparison between the groups of athletes, sedentary controls and sport controls as regards to MNi, NBUDs and binucleated cells frequencies.

Subjects	N	MNi Nab ($\bar{X}$ ± σ)	NBUDs Nab ($\bar{X}$ ± σ)	Binucleated Cells Nab ($\bar{X}$ ± σ)
Athletes	125	25 (0.200 ± 0.402) ^A^	100 (0.800 ± 1.040) ^B^	38 (0.304 ± 0.732) ^C^
Sport Controls	31	34 (1.097 ± 0.908) ^D^	48 (1.548 ± 1.121) ^E^	10 (0.323 ± 0.702)
Sedentary Controls	50	34 (0.680 ± 0.768)	52 (1.040 ± 1.087)	36 (0.720 ± 0.970)

N = Number of studied subjects; $\bar{X}$ = mean; σ = standard deviation; Nab = Number of aberrations; MNi = Micronuclei; NBUDs = Nuclear Buds. ^A^
*p* < 0.001, significantly lower with respect to sedentary and sport controls (Mann–Whitney test); ^B^
*p* < 0.001, significantly lower with respect to sport controls (Mann–Whitney test); ^C^
*p* < 0.001, significantly lower with respect to sedentary controls (Mann–Whitney test); ^D^
*p* = 0.040, significantly higher with respect to sedentary controls (Mann–Whitney test); ^E^
*p* = 0.030, significantly higher with respect to sedentary controls (Mann–Whitney test).

**Table 5 biology-12-01110-t005:** Evaluation of differences in the genomic damage level (MNi and NBUDs) and in binucleated cell frequencies among subjects belonging to different sport disciplines.

Subjects	N	MNi Nab ($\bar{X}$ ± σ)	NBUDs Nab ($\bar{X}$ ± σ)	Binucleated Cells Nab ($\bar{X}$ ± σ)
Martial Arts	34	5 (0.147 ± 0.359)	21 (0.618 ±0.779)	18 (0.529 ± 0.961) ^D^
Basketball	29	4 (0.138 ± 0.351)	16 (0.552 ± 0.736)	5 (0.172 ± 0.468)
Volleyball	40	6 (0.150 ± 0.707)	50 (1.250 ± 3.536) ^B^	15 (0.375 ± 0.000)
Sprinters	22	10 (0.455 ± 0.510) ^A^	13 (0.591 ± 0.666)	0 (0.000 ± 0.000) ^C^

N = Number of studied subjects; Nab = number of aberrations; $\bar{X}$ = mean; σ = standard deviation; MNi = Micronuclei; NBUDs = Nuclear Buds. ^A^
*p* < 0.005, significantly higher with respect to the other three category of athletes (Mann–Whitney test); ^B^
*p* < 0.001, significantly higher with respect to the other three groups (Mann–Whitney test); ^C^
*p* < 0.001, significantly lower with respect to Martial Art and Volleyball groups (Mann–Whitney test); ^D^
*p* = 0.030, significantly higher with respect to the Basketball group (Mann–Whitney test).

**Table 6 biology-12-01110-t006:** Hardy–Weinberg Equilibrium test for the analyzed genotypes.

Gene Polymorphisms	Allele	N	Frequency	Genotype	N	Frequency	HWE *p*-Value	χ^2^-Value
*GSTT*	Positive	--	--	Positive	152	0.738	n.a.	n.a.
	Null	--	--	Null	54	0.262		
*GSTM*	Positive	--	--	Positive	142	0.689	n.a.	n.a.
	Null	--	--	Null	64	0.310		
*CYP1A1*	A	299	0.726	AA	103	0.500	*p* > 0.05	3.701
	G	113	0.274	AG	93	0.451		
				GG	10	0.049		
*XRCC1*	C	295	0.716	CC	100	0.485	*p* > 0.05	3.699
	T	117	0.284	CT	95	0.461		
				TT	11	0.054		
*XPC*	A	282	0.684	AA	98	0.476	*p* > 0.05	0.231
	C	130	0.316	AC	86	0.417		
				CC	22	0.107		

HWE = Hardy–Weinberg Equilibrium; χ^2^ = chi-square test; n.a. = not applicable because the genotyping procedure does not allow to discriminate heterozygotes from dominant homozygotes.

**Table 7 biology-12-01110-t007:** Analysis of the possible association between a specific genotype and the frequencies of MNi, NBUDs and binucleated cells.

Gene	Genotype	N	Total MNi	$\bar{X}$ ± SE	Total NBUDs	$\bar{X}$ ± SE	Total BIN	$\bar{X}$ ± SE
*GSTT1*	Positive	152	66	0.434 ± 0.055	152	1.000 ± 0.080	72	0.474 ± 0.071
	Null	54	27	0.500 ± 0.098	48	0.889 ± 0.142	12	0.222 ± 0.073
*GSTM*	Positive	142	66	0.465 ± 0.059	141	0.993 ± 0.089	60	0.423 ± 0.068
	Null	64	27	0.422 ± 0.083	59	0.922 ± 0.147	24	0.375 ± 0.101
*CYP1A1*	AA	103	53	0.515 ± 0.070	109	1.058 ± 0.119	35	0.340 ± 0.075
	AG	93	36	0.387 ± 0.066	79	0.849 ± 0.093	47	0.505 ± 0.092
	GG	10	4	0.400 ± 0.221	12	1.200 ± 0.389	2	0.200 ± 0.133
*XRCC1*	CC	100	45	0.450 ± 0.067	103	1.030 ± 0.110	39	0.390 ± 0.075
	CT	95	44	0.463 ± 0.073	89	0.937 ± 0.113	44	0.463 ± 0.092
	TT	11	5	0.455 ± 0.027	10	0.909 ± 0.315	3	0.273 ± 0.195
*XPC*	AA	98	41	0.418 ± 0.066	83	0.847 ± 0.100	35	0.357 ± 0.077
	AC	86	41	0.477 ± 0.073	91	1.058 ± 0.123	44	0.512 ± 0.097
	CC	22	11	0.500 ± 0.183	26	1.182 ± 0.276	5	0.227 ± 0.113

N = Number of subjects with a specific genotype; $\bar{X}$ = mean; SE = Standard Error; MNi = Micronuclei; NBUDs = Nuclear Buds; BIN = Binucleated cells.

## Data Availability

The analytical data will be made available to interested parties upon request.

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
