# Peer review of "Regular Sport Activity Is Able to Reduce the Level of Genomic Damage"

_biology, 2023, doi:10.3390/biology12081110_

Round 1

Reviewer 1 Report

Dear Authors, 

The manuscript titled “Regular Sport Activity is able to reduce the level of genomic damage” focuses on the effect of sport activity on DNA damage. The Authors found that different kind of sports are associated with more DNA damage than other sports without gender and gene polymorphisms influences.

Even if the obtained results are fine, I have some concerns and questions about the methods and the data obtained.

1- When the specimens were harvested? This is not specified in the materials and methods. 

2- Did the Authors analysed the levels of cortisol or other hormones in order to evaluate the overtraining status of the subject? This should be an important note especially for occasional athletes. 

3- Did the Authors evaluated also the free DNA? 

4- Concerning table 2, seems that the total female were 58, can the Authors explain why the number of analysed subject with nuclei buds are 64?

5-Also in table 3, if the control subject are 81, 100 studied subject for nuclear buds seems to be too much.

Minor concerns.

1- The table reported in the main manuscript are not aligned with the text and sometime the tables cover the text. Please, fix it.

2- Figure 3, I suppose, seems to show the DNA damage in buccal cells. Please, fix the panels’ letters and indicate the nuclei buds presence the seems to be in panel D. 

3-Please, use the same colo all over the text, avoiding red for statistical significane symbols.

Author Response

Reviewer 1

Dear Reviewer,

Thank you for your precious revision of our paper. Below you can find the answers to each single point.

In the revised version of the paper.

  1. When the specimens were harvested? This is not specified in the materials and methods. 

Authors: in the revised version, we specified the sampling period. We added the following information: “sampling began in November 2021 and ended in June 2022”.

  1. Did the Authors analysed the levels of cortisol or other hormones in order to evaluate the overtraining status of the subject? This should be an important note especially for occasional

Authors: No, we didn’t. Although it is known that physical activity can also represent a “stress factor” and that some hormones, such as testosterone and cortisol, can be used to assess the metabolic alterations caused by exercise, we decided to focus the object of our work on the genomic damage. The study required a long research time, i.e. more than 2 years. Please, note that, among studies applying the Micronucleus assay, our has one of the largest samples ever. Considering another physiological parameter would have required further time dilation.

Moreover, as evidenced in a recent published paper, there is a variability in individual hormones response. Indeed, although testosterone and cortisol concentrations, as well as the testosterone-cortisol ratio are sensitive to changes in athlete load, the direction of their response is often inconsistent and is likely influenced by player training status and non-sport-related stressors (Springham et al., 2022).

Springham, M.;  Newton R.U.;  Strudwick, A.J.; Waldron, M. Selected Immunoendocrine Measures for Monitoring Responses to Training and Match Load in Professional Association Football: A Review of the Evidence. Int. J. Sports Physiol. Perform. 2022, 17(12):1654-1663.

  1. Did the Authors evaluated also the free DNA? 

Authors: Also in this case, and for the same motifs explained to the point 2, the answer is: No, we didn’t. It is known that strenuous physical exercise causes a massive elevation in the concentration of circulating cell-free DNA (cfDNA), correlated with effort intensity and duration. Increases of cfDNA due to exercise represent a potential hallmark for the overtraining syndrome and might have relevance in clinical medicine (Breitbach et al., 2012). However, our sample of athletes is mostly constituted by non-elite athletes that don’t reach the condition of overtraining, as they typically train 3-6 times/week. Vice versa, this condition is typical of elite athletes that train 10-12 times/week and more. Moreover, it is also known that the cfDNA concentrations peak immediately after acute exercise but rapidly return to baseline (Breitbach et al., 2012). We sampled the subjects before their training session and, therefore, most likely we would not have seen high levels of cfDNA.

Breitbach, S., Tug, S., Simon, P. Circulating Cell-Free DNA. An Up-Coming Molecular Marker in Exercise Physiology. Sports Med. 2012, 42, 565–586.

  1. Concerning table 2, seems that the total female were 58, can the Authors explain why the number of analysed subject with nuclei buds are 64? Also in table 3, if the control subject are 81, 100 studied subject for nuclear buds seems to be too much.

Authors: We apologise for the ambiguity, the letter N can be confusing. The N under MNi, NBUDs and Binucleated Cells columns indicates the number of observed aberrations in the total number of the subjects and not the number of subjects with that specific aberration. However, we realized that this N could be confused with that of the second column, which refers to the number of subjects sampled. In the revised version, from the third column of the tables, we substituted N with Nab (= number of aberrations).

Minor concerns.

  1. The table reported in the main manuscript are not aligned with the text and sometime the tables cover the text. Please, fix it.

Authors: we apologize. In the revised version, we aligned tables and texts.

  1. Figure 3, I suppose, seems to show the DNA damage in buccal cells. Please, fix the panels’ letters and indicate the nuclei buds presence the seems to be in panel D. 

Authors: we apologize. In the revised version, we made the panel’s letters visible and MNi and NBUDs are evidenced with arrows.

  1. Please, use the same color all over the text, avoiding red for statistical significane symbols.

Authors: We have uniformed the color in the whole text, tables included.

Reviewer 2 Report

The research topic is an actual problem. The article is devoted to the study of the influence of the intensity of physical activity on genomic and nuclear damage. Four groups of athletes were studied: sprinters, martial artists, basketball players and volleyball players. The study was conducted in two stages, at the first stage groups were compared according to the intensity of physical activity, at the second stage - by sports.

The authors describe very well the methods of collecting and processing the DNA of the test subjects. Polymerase chain reaction (PCR) was performed for CYP1A1, GSTM1, GSTT1, XRCC1, XPC. Statistical analysis was performed using the SPSS program using methods: Spearman correlation and the Man-Whitney test. The originality of the text is: 64.13%.

Despite the completeness of the presentation of the material, the following remarks are made:

- In the "Introduction" section, the analysis of previous studies on this topic is carried out very poorly.

- Please send a copy of the protocol of the University of Turin (protocol number 0609375, 28.10.2021) on research ethics.

- In statistical analysis, it is necessary to use the Hardy-Weinberg method for each of the polymorphisms. The Shapiro-Wilkie method is not listed in Section 2.4 and needs to be described in detail.

- It is necessary to use the method of associations instead of the method of correlations.

- Statistical descriptions: arithmetic mean and error of the arithmetic mean can be improved in the description by using special characters.

- It is necessary to add the research hypothesis at the end of section "I Introduction".

- It is necessary to add a reference in line 11 after the first sentence.

- In my opinion, there is no direct evidence in the article about the effect of low-intensity physical activity on DNA damage.

- It is not clear from the work what is genomic damage? How was it assessed, which polymorphisms were damaged and how? The organization of work should be described in more detail and clearly.

- In line 285, the imposition of a picture on the text.

- The “Simple Summary:” section is usually not written.

The article can be recommended for publication after the correction of these comments.

Author Response

Reviewer 2

Dear Reviewer,

Thank you for your precious revision of our paper. Below you can find the answers to each single point.

In the revised version of the paper.

Introduction:

  1. In the introduction, the authors generally describe the conditions for the formation of free radicals and their impact on the athlete's body. However, special attention should be paid to the main factors inducing oxidative DNA damage (Reactive oxygen species, hyperironemia, etc.).

Authors: as suggested, we improved the Introduction session adding the following paragraphs:

Although it is known that exercise improves health and has a protective effects against diseases, it increases the production of reactive oxygen species (ROS) and free radicals, as part of the muscular adaptation to training process (Zanella et al., 2019).

Besides, varying concentration of some metals as iron and copper can generate radicals as well (Meneghini, 1997). From a genomic point of view, excessive ROS production is associated with the oxidative DNA damage and cancer progression (Renaudin et al., 2021). This can happen, for example, because chromatine-bound iron reacts with environmental H2O2 producing OH radical, a very reactive species that affects nearby DNA (Meneghini, 1997).

  • Zanella, P.B.; August, P.M.; Alves, F.D.; Matté, C.; de Souza, C.G. Association of Healthy Eating Index and oxidative stress in adolescent volleyball athletes and non-athletes. Nutrition 2019, 60, 230-234.
  • Meneghini, R. Iron homeostasis, oxidative stress, and DNA damage. Free Radic. Biol. Med. 1997, 23, 783-792.
  • Renaudin, X. Reactive oxygen species and DNA damage response in cancer. Rev. Cell. Mol. Biol. 2021, 364, 139-161.

  1. Morover in order to assess the distribution of body weight in the population and which people were included in isi urgent to add information how it was measured.

Authors: In order to determine physical activity-related and physiological variables, a questionnaire was provided to all study participants. Weight data were recorded on the basis of what was indicated in the questionnaire by each subject participating in the study. However, we would like to emphasize that all athletes, before starting the training session, measured their weight and, therefore, the data they provided can be considered highly realistic.For the seek of homogeneity, we adopted the same criteria also for the control subjects.

Materials and Methods

  1. Was the sampe size calculated for the statistical significance of the obtained results????

Authors: as suggested, we performed the power analysis and the results are listed below. As you can see, the power of our sample is very high (0,99) for all categories and biomarkers, with exception of binucleated cells for “Sport Controls” group, showing a power value of 0.65. We included these data in the result session.

POWER ANALYSIS

Athletes N = 125

N

Effective Power

Power

St. Dev.

Size Effect

Sign.

MNi

77

0.991

0.99

0.402

0.498

0.05

NBUDs

34

0.992

0.99

1.04

0.769

0.05

Binucleated Cells

109

0.990

0.99

0.732

0.415

0.05

Controls N = 81

N

Effective Power

Power

St. Dev.

Size Effect

Sign.

MNi

21

0.991

0.99

0.843

0.996

0.05

NBUDs

18

0.993

0.99

1.121

1.102

0.05

Binucleated Cells

48

0.991

0.99

0.894

0.635

0.05

Sport Controls N = 31

N

Effective Power

Power

St. Dev.

Size Effect

Sign.

MNi

15

0.991

0.99

0.908

1.208

0.05

NBUDs

12

0.991

0.99

1.121

1.381

0.05

Binucleated Cells

30

0.661

0.65

0.720

0.449

0.05

Sedentary Controls N = 50

N

Effective Power

Power

St. Dev.

Size Effect

Sign.

MNi

26

0.991

0.99

0.768

0.885

0.05

NBUDs

23

0.992

0.99

1.087

0.957

0.05

Binucleated Cells

36

0.991

0.99

0.970

0.742

0.05

  1. Authors write that they included in the study population of 81 control subjects - what was the main criteria for the sedentary controls - how it ws measured ???? And same situation for the sports controls. If we write subjects who practice sports occasionally, and no more than two times a week important factor will be the duration of the exercises and its type...

Authors: as suggested, we improved the text, better explaining, in the materials and methods session; what do we mean by “sedentary controls” and “sports controls”. In particular, we added the following data:

  • Sedentary controls, are individuals who do not practice any sporting activity, not even occasionally.
  • Sport controls, are subjects who train no more than 2 times a week for no more than 2 hours per training.

  1. How many people which did not meat criteria of the study was excluded from the sample? 

Authors: we excluded from the sample smokers and individuals who reported alcohol consumption, treatment for acute infections and/or chronic non-infectious diseases, history of cancer and exposure to diagnostic X-rays, for at least one year prior the analysis, for a total of 220 subjects. This datum was included in the revised version of the paper.

  1. Please add information on energy requirements and metabolic orientation in the subpopulations listed in the study.

Authors: We apologize but we didn’t record data about energy requirements. However, we         would like to emphasize that we decided to focus the objective of the research work on the genomic damage. Our study required a long research time (2 years), and considering another physiological parameter would have required further time dilation. Vice versa, in result session, we specified the different metabolic orientation associated to the subpopulations. In particular, Martial Arts, Basket and Volley are considered intermittent metabolic activities, i.e. sports where aerobic activity alternates with anaerobic activity and breaks. Vice versa, sprint activity is considered an anaerobic activity. For all sports, the training sessions were 1.5 hours long for 3-6 times/week”.

  1. Please elaborate on the statistical methods used in the study, whether they meet all the criteria. 

Authors: We are not sure about what the reviewer required. However, we improved the                statistical analyses session, also including the power analysis. Moreover, we repeated the statistical elaborations and the results perfectly matched the previous ones.

Results:

  1. Ad information about weight and BMI in order to assess the distribution of body weight in the population and which people were included in the study (Table. 1).

Authors: In Table 1 data about weight was already included, as well as the results of statistical analyses. Unfortunately, we were unable to include data about the BMI because the data provided by the interviewed subjects were not complete, with lack of information about height in some cases and unclear in others. We asked the sampled subjects to fill in the questionnaire before the training sessions; in some cases they forgot to provide information (or decided to avoid it) on height or other data. For this reason, we decided to exclude from table 1 the incomplete data about height and, consequently, those about BMI. However, we would like to emphasise that all sports subjects tend to have, by virtue of the sporting activity practiced, a relatively healthy lifestyle, in terms of diet, smoking and alcohol consumption, making the samples very homogeneous.

Discussion

  1. Please expand on the practical value of the above considerations and how it relates to the level of proteins synthesized during metabolism.

Authors: we expanded the discussion session with the following paragraph:

Moreover, we would like to emphasize that sport activity is able not only to reduce genomic damage but is a potent stimulators of muscle protein synthesis: it was demonstrated that the levels of muscle protein synthesis after an acute exercise are modulated by an individual’s training status, whereas stress induced by an unaccustomed resistence exercise, typical, for example, of subjects who practice sports only occasionally, could result in muscle damage  (Damas et al., 2015). 

Damas, F., Phillips, S., Cassaro Vechin, F., Ugrinowitsch, C. A Review of Resistance Training-Induced Changes in Skeletal Muscle Protein Synthesis and Their Contribution to Hypertrophy. Sports Med. 2015, 45, 801–807

  1. Comments on the Quality of English Language

I have observed minor editing errors - some English language correction would be required, but article is welll writen. 

                Authors: the paper was revised by a native English speaker who made some corrections

Reviewer 3 Report

Authors present interesting article about the influance of regular sport activity in reduction of the the level of genomic damage. Work is well writen and interesting for the reader but there are som concers that should be corrected in order to preapare presented work for the publication.  

Introduction:

In the introduction, the authors generally describe the conditions for the formation of free radicals and their impact on the athlete's body. However, special attention should be paid to the main factors inducing oxidative DNA damage (Reactive oxygen species, hyperironemia, etc.). Morover in order to assess the distribution of body weight in the population and which people were included in isi urgent to add information how it was messured.

Materials and Methods

Was the sampe size calculated for the statistical significante of the obtained results????

Authors write that they included in the study population of 81 control subjects - what was the main criteria for the sedentary controls - how it ws messured ???? And same situation for the sports controls. If we write subjects who practice sports occasionally, and no more than two times a week inmprtaint factor will be the duration of the exercises and its type...

How many people which did not meat criteria of the study was excluded from the sample? 

Please add information on energy requirements and metabolic orientation in the subpopulations listed in the study.

Please elaborate on the statistical methods used in the study, whether they meet all the criteria. 

Results:

Ad information about weight and BMI in order to assess the distribution of body weight in the population and which people were included in the study (Table. 1).

Disscusion

Please expand on the practical value of the above considerations and how it relates to the level of proteins synthesized during metabolism.

I have observed minor editing errors - some English language correction would be required, but articel is welll writen. 

Author Response

Dear Reviewer,

Thank you for your precious revision of our paper. Below you can find the answers to each single point.

In the revised version of the paper.

Comments and Suggestions for Authors:

“……….”  

Despite the completeness of the presentation of the material, the following remarks are made:

  1. In the "Introduction" section, the analysis of previous studies on this topic is carried out very poorly.

Authors: as suggested, in the improved version we included previous studies on the topic of the paper. In particular, we added the following paragraph highlighting the results of the few published works about the relationship between micronuclei and sport activity:

“Research examining the impact of sport activity on the frequency of MNi and other aberrations is limited, and usually relies on relatively small samples sizes [20-22]. While some studies revealed a correlation between phyisical activity and an increase in MNi frequency [21], particularly when intense [20], others did not found a significant effect [22]. It is essential to conduct more extensive investigations in this area to better understand the possible relationship between sport activity and these biomarkers”.

  1. Pittaluga, M.; Parisi, P.; Sabatini, S.; Ceci, R.; Caporossi, D.; Catani, M.V.; Savini, I.; Avigliano, L. Cellular and biochemical parameters of exercise-induced oxidative stress: relationship with training levels. Free radic. Res. 2006, 40(6), 607–614.
  2. Sharma, R.; Shailey; Gandhi, G. Pre-cancerous (DNA and chromosomal) lesions in professional sports. Canc. Res. Ther. 2012, 8(4), 578-585.
  3. Franzke, B.; Schober-Halper, B.; Hofmann, M.; Oesen, S.; Tosevska, A.; Nersesyan, A.; Knasmüller, S.; Strasser, E.M.; Wallner, M.; Wessner, B.; Wagner, K.H. Chromosomal stability in buccal cells was linked to age but not affected by exercise and nutrients – Vienna Active Ageing Study (VAAS), a randomized controlled trial. Redox Biol. 2020, 28, 101362.

  1. Please send a copy of the protocol of the University of Turin (protocol number 0609375, 28.10.2021) on research ethics.

Authors: in the revised version of the paper we attached a copy of the protocol provided by the University of Turin.

  1. In statistical analysis, it is necessary to use the Hardy-Weinberg method for each of the polymorphisms.

Authors: In the revised version, we included a tale (Table n. 6) with the HWE analysis. As you can see, all gene polymorphisms were in HWE.

  1. The Shapiro-Wilkie method is not listed in Section 2.4 and needs to be described in detail.

Authors: in the revised version, in section 2.4, we included detailed description of the Shapiro-Wilk test.                                                                                                           

  1. It is necessary to use the method of associations instead of the method of correlations.

Authors: in the revised version of the paper, we provided a table showing the association analysis (Table 7). In the whole sample, no association was found between gene polymorphisms and the frequency of MNi, NBUDs and Binucleated Cells.

  1. Statistical descriptions: arithmetic mean and error of the arithmetic mean can be improved in the description by using special characters.

Authors: in the revised version of the paper, we used special characters in order to describe mean (μ), standard deviation (σ) and standard error (SE).

  1. It is necessary to add the research hypothesis at the end of section "I Introduction".

Authors: as suggested, we schematically described our research hypothesis at the end of the Introduction session.

  1. It is necessary to add a reference in line 11 after the first sentence.

Authors: line 11 is included in Summary, that does not require references.

  1. In my opinion, there is no direct evidence in the article about the effect of low-intensity physical activity on DNA damage.

Authors: we agree with your consideration. In the discussion session of the revised version, we better specified that the discriminating factor, in terms of genomic damage, is the constancy of the physical activity with respect to a sport activity practiced occasionally.

  1. It is not clear from the work what is genomic damage? How was it assessed, which polymorphisms were damaged and how? The organization of work should be described in more detail and clearly.

Authors: in the revised version of the paper, we better specified what we intended for “genomic damage” (MNi, NBUds) and in many cases we substituted the terms “genomic damage” with the name of the specific markers analysed. With regard to gene polymorphisms, in the introduction session, we better specified the role of the analysed polymorphisms in modulating the frequencies of cytogenetic markers.

  1. In line 285, the imposition of a picture on the text.

Authors: we apologize. In the revision version of the paper, we better aligned tables and figures.

  1. The “Simple Summary:” section is usually not written.

Authors: In the model provided us by the Editor, the summary was required. Moreover, we observed the last issues of the journal and the summary was present in all accepted papers. However, if for the Editor the summary is not necessary, we will remove it.

The article can be recommended for publication after the correction of these comments.

Authors: We would like to thank the anonymous referee for the suggestions that helped us to improve the paper.

Round 2

Reviewer 1 Report

Dear Authors, 

Many thanks for your replay and for following the reviewers suggestion. The quality of the paper has been improved and should be considered for publication.

Author Response

Reviewer 1: “Dear Authors, Many thanks for your replay and for following the reviewers suggestion. The quality of the paper has been improved and should be considered for publication”.

Authors: We thank the anonymous reviewer, whose suggestions improved the paper. We also thank the reviewer for his positive consideration of our work.

Reviewer 2 Report

Dear authors!

 1. In table 6, in the column "HWE P-value" you need to indicate "P> 0.05". The numbers you provided are incomprehensible.

 2. You do not correctly designate the "Arithmetic Mean" indicator. The symbol you are using stands for "Probability Mean". Need to replace it. The arithmetic mean is usually denoted as X with a horizontal bar on top.

Author Response

Reviwer 2:

  1. In table 6, in the column "HWE P-value" you need to indicate "P> 0.05". The numbers you provided are incomprehensible.

Authors: as suggested, in Table 6, we indicate the HWE P-values as P>0.05

  1. You do not correctly designate the "Arithmetic Mean" indicator. The symbol you are using stands for "Probability Mean". Need to replace it. The arithmetic mean is usually denoted as X with a horizontal bar on top.

Authors: as suggested, in the revised version of the paper we indicated the arithmetic mean as X with a horizontal bar on top.

Reviewer 3 Report

In my opinion, the applied corrections significantly increased the value of the presented work, which allows to proceed with further stages of its publication

Author Response

Reviewer 3: “In my opinion, the applied corrections significantly increased the value of the presented work, which allows to proceed with further stages of its publication”.

Authors: We thank the anonymous reviewer, whose suggestions improved the paper. We also thank the reviewer for his positive consideration of our work.